# Dietary Vitamin E and/or Hydroxytyrosol Supplementation to Sows during Late Pregnancy and Lactation Modifies the Lipid Composition of Colostrum and Milk

**DOI:** 10.3390/antiox12051039

**Published:** 2023-05-03

**Authors:** Hernan D. Laviano, Gerardo Gómez, María Muñoz, Juan M. García-Casco, Yolanda Nuñez, Rosa Escudero, Ana Heras Molina, Antonio González-Bulnes, Cristina Óvilo, Clemente López-Bote, Ana I. Rey

**Affiliations:** 1Departamento Producción Animal, Facultad de Veterinaria, Universidad Complutense de Madrid, Avda. Puerta de Hierro s/n., 28040 Madrid, Spain; hernanlavianomvz03@gmail.com (H.D.L.); rmescude@ucm.es (R.E.); andelash@ucm.es (A.H.M.); clemente@ucm.es (C.L.-B.); 2Instituto Regional de Investigación y Desarrollo Agroalimentario y Forestal de Castilla-La Mancha (IRIAF), 13700 Toledo, Spain; g.gomez.mat@gmail.com; 3Departamento de Mejora Genética Animal, Instituto Nacional de Investigación y Tecnología Agraria y Alimentaria (INIA), Consejo Superior de Investigaciones Científicas (CSIC), Ctra Coruña km 7.5, 28040 Madrid, Spain; mariamm@inia.csic.es (M.M.); garcia.juan@inia.csic.es (J.M.G.-C.); nunez.yolanda@inia.csic.es (Y.N.); ovilo@inia.csic.es (C.Ó.); 4Departamento de Producción y Sanidad Animal, Facultad de Veterinaria, Universidad Cardenal Herrera-CEU, CEU Universities, C/ Tirant lo Blanc, 7, Alfara del Patriarca, 46115 Valencia, Spain; antonio.gonzalezbulnes@uchceu.es

**Keywords:** sow’s lactation, colostrum composition, milk composition, vitamin E, hydroxytyrosol

## Abstract

Modifying the composition of a sow’s milk could be a strategy to improve the intestinal health and growth of her piglet during the first weeks of life. This study evaluated how dietary supplementation of vitamin E (VE), hydroxytyrosol (HXT) or VE+HXT given to Iberian sows from late gestation affected the colostrum and milk composition, lipid stability and their relationship with the piglet’s oxidative status. Colostrum from VE-supplemented sows had greater C18:1n−7 than non-supplemented sows, whereas HXT increased polyunsaturated (∑PUFAs), ∑n−6 and ∑n−3 fatty acids. In 7-day milk, the main effects were induced by VE supplementation that decreased ∑PUFAs, ∑n−6 and ∑n−3 and increased the Δ-6-desaturase activity. The VE+HXT supplementation resulted in lower desaturase capacity in 20-day milk. Positive correlations were observed between the estimated mean milk energy output and the desaturation capacity of sows. The lowest concentration of malondialdehyde (MDA) in milk was observed in VE-supplemented groups, whereas HXT supplementation increased oxidation. Milk lipid oxidation was negatively correlated with the sow’s plasma oxidative status and to a great extent with the oxidative status of piglets after weaning. Maternal VE supplementation produced a more beneficial milk composition to improve the oxidative status of piglets, which could promote gut health and piglet growth during the first weeks, but more research is needed to clarify this.

## 1. Introduction

Milk is a vital food to guarantee the survival of mammalian animals during the first weeks of life. In the case of a piglet that is born without hair and with scarce energy reserves, milk is the main energy vehicle to support thermogenesis [1]. Thus, sow milk is characterized by containing, compared with other non-ruminant and ruminant species, a greater proportion of protein, fat and lactose [2,3]. However, milk is not only a nutrient-rich fluid suitable for growth but its composition can enhance intestinal mucosa development and promote the growth of certain bacteria in the intestine or limit the growth of others, thus affecting intestinal health [4].

Milk components, mainly those that are not synthetized in the mammary gland, can be modified by factors such as feeding [5]. Hence, some previous research on the effects of liposoluble micronutrients, such as vitamin E (VE), on milk composition found that when administered at high doses during the gestation and lactation periods, it increased its concentration in colostrum and milk and the fat component, and thus, affected the health status and growth of piglets [6]. This could be a relevant aspect in the case of the Iberian pig, which was shown to have lower growth rates than those of improved genotypes, partly due to the limited use of energy and milk protein [3]. The health and productivity effect of VE-supplementation in piglets [6] was attributed to their potent chain-breaking antioxidant effect on cells [7], which could also protect the epithelial barrier function [8]. It is widely documented that supplementation with high doses of VE is an effective strategy to avoid the important decline of VE plasma concentration post-weaning [9,10,11] and may help to counteract the oxidative state of piglets in this critical period [6,9,10,11]. Moreover, a study that investigated the effects of supplementation with VE (150 mg/kg) in its different chemical forms on the fatty acid profile of sow’s milk showed a positive effect on the proportion of some specific fatty acids [12]. Fatty acids may participate in the oxidative balance of the body to some extent by promoting or inhibiting lipid oxidation [13], and they can also be used with different efficiencies for energy production [14] or affect gut health [8]. Therefore, the study of how fatty acids or other milk components can be modified by antioxidant micronutrient supplementation deserves further attention.

The effect of supplementing diets with phenolic hydrophilic antioxidants on milk composition was also investigated in other species. In lactating buffaloes, dried stoned olive pomace supplementation was reported to improve milk tocopherols and retinol [15]. Other antioxidants derived from olive leaves, such as oleuropein or hydroxytyrosol, were shown to not only have powerful antioxidant effects but also to produce changes in the metabolic use of fatty acids [16], as well as hypolipidemic and hypoglycemic effects [17]. However, the effect of dietary supplementation on the composition of milk has not been investigated. There is also a lack of information on the possible combined effects of different antioxidants (lipophilic and non-lipophilic, such as polyphenols) on sow’s milk and colostrum composition. Considering that the growth and gut health of nursery piglets depend largely on the quantity and quality of milk production [18,19], extending knowledge on how different antioxidants supplementation in diets or their combined administration affects colostrum and milk composition is a matter of interest.

Furthermore, the post-weaning period is a major challenge for a piglet when trying to maintain homeostasis by having to deal with numerous pro-oxidant agents [9,10,11]. This control of the oxidative stress to which the animal is subjected during the first days of life plays an important role in the metabolism and health of the adult due to its effects on the development of inflammation and the immune response [20]. On many occasions, the extra contribution of antioxidants in order to achieve antioxidant/pro-oxidant balance must be provided by food, including breast milk [9,10,11]. In this sense, it is unknown how HXT can modify the oxidative stability of milk compared with the use of VE.

It is hypothesized that VE, HXT or their combined administration to sows from gestation could modify the composition and lipid stability of milk in different ways and then affect the oxidative status.

Thus, the objective of the present research was to study the effects of dietary supplementation of VE (100 mg/kg), hydroxytyrosol (HXT) (1.5 mg/kg) or a combined administration provided to Iberian sows during late gestation and lactation on the colostrum and milk composition, lipid stability and their possible relationships with piglets’ oxidative status.

## 2. Materials and Methods

### 2.1. Chemicals

All chemicals used were analytical grade and were supplied by Sigma-Aldrich (Alcobendas, Madrid, Spain), Panreac (Castellar del Vallès, Barcelona, Spain) or Scharlau (Sentmenat, Barcelona, Spain).

### 2.2. Animals, Experimental Diets and Sample Collection

The experimental procedures used in this study were in compliance with the Spanish guidelines (RD53/2013) [21] and European Union Directive 2010/63/UE for the care and use of animals in research [22]. The experimental procedures (report ORCEEA 2019-10) were approved by the INIA Committee of Ethics in Animal Research.

The study was carried out at the facilities of Dehesón del Encinar (Oropesa, Toledo, Spain). Fifty pregnant Iberian sows (half primiparous and half multiparous with between 4 and 5 parity) were divided into four experimental groups (each group with equal distribution of primiparous and multiparous), according to the dietary supplementation from day 85 of gestation (average weight of 126.2 ± 29.3 kg) until weaning (28 days): (1) the control group received a 30 mg supplementation of α-tocopheryl acetate/kg feed, (2) the VE group received 100 mg of α-tocopheryl acetate/kg, (3) the HXT group received 30 mg α-tocopheryl acetate/kg and 1.5 mg hydroxytyrosol/kg, and (4) the VE+HXT group received 100 mg of α-tocopheryl acetate/kg + 1.5 mg hydroxytyrosol/kg feed. The basal level of α-tocopherol in the feed ingredients was 12–14 mg/kg; thus, the total vitamin E concentration in all groups was in compliance with values recommended for reproductive sows [23]. The highest VE dose of 100 mg/kg was used after taking into account the effective antioxidant effect observed in a previous study [16]. During this experimental period, the feed administration was adjusted to fulfill daily requirements according to the National Research Council [23] (Table A1, Appendix A), and water was provided ad libitum. In the pre-experimental period (from the end of natural service to day 85 of pregnancy), sows were given a standard grain-based diet (g/kg: 888 dry matter, 124.6 protein, 29.9 fat, 49.3 fiber, 62.1 ash; 3050 kcal/kg metabolic energy) (Sanchez Romero Carvajal, Jabugo, Spain). The α-tocopheryl acetate used in the diets was purchased from DSM Nutritional Products (Alcalá de Henares, Madrid, Spain) and the hydroxytyrosol extract (*Olea europaea* L. dry extract, N20130102) was obtained from Natac (Alcorcón, Madrid, Spain).

Milk and colostrum (5 mL) samples were taken from a representative number of multiparous sows (n = 7 per treatment) with a homogeneous litter size on the day of delivery and 7 and 20 days post-partum. Piglets were separated from their mother and milk was collected from the functional glands by hand-milking after disinfecting the operator’s hands and nipples with soap and water, and put into plastic tubes of 20 mL. Samples were kept under refrigeration while handling, and then immediately frozen at −20 °C until analysis (within 2 months after collection). The mean milk energy output (kcal/day) was estimated according to the following equation: 4.92× mean litter gain (g/day) − (90× litter size) [22].

Blood samples from sows for oxidative status analysis were obtained coinciding with the taking of milk samples (at the peak of lactation on day 20 after farrowing), which is the moment when the production is at a maximum and composition could be affected [5,24]. In addition, blood was taken from piglets (1 male piglet was selected from each litter at weaning). Blood was collected in sterile EDTA vacuum tubes (Vacutainer, BD, Franklin Lakes, NJ, USA), immediately centrifuged at 2500 rpm for 10 min and plasma samples were kept at −80 °C until analysis (less than 1 month).

### 2.3. Laboratory Analysis

#### 2.3.1. Lactose Concentrations of the Colostrum and Milk

Lactose was measured according to Beutler [25] with few modifications. For the sample preparation, milk or colostrum (100 µL) was placed in a 1.5 mL Eppendorf tube in the presence of distilled water. The mixture was heated in a stirring bath for 15 min at 50 °C. For sample clarification, potassium hexacyanoferrate (Carrez I solution), zinc sulfate (Carrez II solution) and NaOH (100 mM) were then added in a total volume of 1 mL, mixed and centrifuged at 10,000 r.p.m. (Hermle Z383-K, Wehingen, Germany). The intermediate phase was collected for lactose determination. Lactose was determined via the addition of β-galactosidase after shaking and incubating the sample extract at 25 °C for 10 min in the presence of a tamponed solution (pH 8.6), NAD+ and β-galactose dehydrogenase. The lactose concentration was measured via the increase in absorbance at 340 nm according to the commercial kit procedure (K-Lacgar 05/17, Megazyme, Scotland, UK)

#### 2.3.2. Fat Quantifications of the Colostrum and Milk

Fat was quantified according to a modification of the Rose–Gottlieb method (905.02) [26]. Milk was placed in the presence of NAOH (2.5%), and fat was extracted via the addition of hexane after centrifugation (Hermle Z383-K, Wehingen, Germany). The superior phase was then collected and evaporated under a nitrogen stream and the residue was weighted for fat quantification.

#### 2.3.3. Tocopherol and Retinol Quantifications of the Colostrum and Milk

The α-tocopherol and retinol concentrations in colostrum and milk samples were quantified as described by Rey et al. [27]. Tocopherols were extracted in duplicate samples via saponification in the presence of KOH (50%) and analyzed via reverse phase HPLC (HP 1200, equipped with a diode array detector) (Agilent Technologies, Waldbronn, Germany). The mobile phase was methanol:water (97:3 *v*/*v*) at a flow rate of 2 mL/min and the diode array detector was set at 292 nm. Separation was performed on a LiChrospher 100 RP-18 column (250-4 column size, 5 μm particle size) (Agilent Technologies GmbH, Waldbronn, Germany). Identification and quantification were carried out by means of a standard curve (R^2^ = 0.999) built using the pure compound (Sigma, Alcobendas, Madrid, Spain).

#### 2.3.4. Fatty Acid Compositions of the Colostrum and Milk

Lipids from colostrum and milk were determined via the one-step procedure proposed by Sukhija and Palmquist [28] with modifications. A lyophilized sample (200 mg) was placed in a test tube and toluene (1 mL) containing internal standard (10 mg C15:0/mL) (Sigma-Aldrich, Tres Cantos, Madrid, Spain) and 3 mL of freshly made 10% methanolic-acetyl chloride solution were added. The tubes were vortexed and heated in a shaking water bath at 70 °C for 2 h. Then, potassium carbonate (5%) and toluene were added, mixed and centrifuged at 600× *g* for 5 min. The superior layer containing the fatty acid methyl esters was collected for analysis. Fatty acids were identified and quantified via gas chromatography as described by Rey et al. [12] using a 6890 Hewlett Packard (Avondale, PA, USA) gas chromatograph equipped with an automatic injector maintained at 250 °C and a flame ionization detector. The fatty acid methyl esters were separated in a capillary column (HP-Innowax, 30 m × 0.32 mm id and 0.25 μm cross-linked polyethylene glycol) (Agilent Technologies GmbH, Waldbronn, Germany) using a temperature program of 170 to 245 °C. A split ratio of 1:50 was used. Identification of each fatty acid was carried out via the use of the mixtures of known standards (Sigma-Aldrich, Tres Cantos, Madrid, Spain). Fatty acids were expressed as a percentage of the total fatty acids.

Different indices were measured to estimate desaturase or elongase activities [29,30,31]:

The Δ-9 − desaturase index was calculated as the ratio of the monounsaturated fatty acid to the sum of the monounsaturated fatty acid plus the saturated fatty acid of the same number of carbons using the following equation [29,30]: Δ-9 − desaturase index = (C14:1 n−5 + C16:1 n−7 + C18:1n−9)/(C14:0 + C14:1 n−5+ C16:0 + C16:1 n−7 + C18:0 + C18:1n−9).

The Δ-5 and Δ-6−desaturase indices are meant for the evaluation of the enzymes that participate in the desaturation of C18:2 n−6 and C18:3 n−3 to their long-chain fatty acids and were calculated with the following equations [30,31]:Δ-5-desaturase = (C20:4 n−6)/(C20:3 n−6 + C20:4 n−6)
Δ-6-desaturase = (C18:3 n−6 + C18:4 n−3)/(C18:2 n−6 + C18:3 n−3 + C18:3 n−6 + C18:4 n−3)

The elongase indices were calculated as the ratios of C18:0 to C16:0 and C20:0 to C18:0, whereas the thioesterase index was calculated as the ratio of C16:0 to C14:0 [29,31].

#### 2.3.5. Oxidative Statuses of the Milk Samples, Sows and Piglets

The susceptibility of milk homogenates to iron-induced lipid oxidation was determined as described elsewhere [32] with few modifications. Briefly, samples were incubated at 37 °C in the presence of a Tris–malate buffer (pH 7.4) and ascorbic acid in a total volume of 5 mL. To start the lipid oxidation, 1 mM FeSO_4_ was added to homogenates. At fixed time intervals (0, 30, 60 and 90 min), 1 mL aliquots were removed, mixed with TBA-TCA-HCl reagent in 1:2 proportion and after heating, extract containing thiobarbituric acid-reactive substances (TBARs) were measured spectrophotometrically (532 nm) (ScanGo, Thermofisher Scientific, Alcobendas, Spain). TBARs were expressed as nM malondialdehyde (MDA)/mg protein. The protein content was measured via the procedure of Bradford [33].

The oxidative status of plasma samples from sows (catalase enzyme activity and α-tocopherol) and piglets (MDA) was evaluated spectrophotometrically (Multiscan ScanGo, Thermo-Fisher Scientific, Alcobendas, Spain) as described elsewhere [10,11,16]. Analysis of catalase was carried out according to the kit’s instructions (Arbor Assays, Ann Arbor, MI, USA). Sow’s plasma (50 µL) was diluted with an assay buffer (1:10) and these dilutions (25 µL) were mixed with 25 µL of the hydrogen peroxide reagent (H_2_O_2_), 25 µL of the substrate solution and 25 µL of horseradish peroxidase (HRP), and incubated for 15 min at room temperature. The HRP reacted in the presence of H_2_O_2_ to convert the colorless substrate into a pink-colored product that was read spectrophotometrically at 560 nm. The catalase activity was expressed as U/mL.

### 2.4. Statistical Analysis

The experimental unit for analysis of all data was the individual sow. Data were analyzed following a completely randomized design using the general linear model (GLM) procedure contained in SAS (version 9.4; SAS Inst. Inc., Cary, NC, USA) that included the fixed effects of VE and HXT supplementation and their interaction in a factorial model. A comparative analysis between means was conducted using the Duncan test. Data are presented as the mean of each group and the root-mean-square error (RMSE), together with the significance levels (*p*-value) of the main effects and interaction. The relationship between the mean milk energy output and the C16:1/C16:0 or C18:1/C18: ratios, milk MDA concentration and milk fatty acid proportions, milk MDA and sow’s plasma oxidative stability or piglet’s oxidative status were quantified with regression equations using Statgraphics-19. Differences between means were considered statistically significant at *p* < 0.05 and *p*-values between 0.05 and 0.10 were considered a trend.

## 3. Results

### 3.1. General Compositions of the Colostrum and Milk

The composition of colostrum and milk as affected by VE or HXT supplementation is shown in Table 1. Neither the overall composition of milk nor colostrum (dry matter, lactose, protein, fat) was affected by the dietary antioxidant supplementation of sows. However, dietary VE supplementation tended (*p* = 0.090) to increase the lactose percentage of colostrum and decrease the protein percentage of milk on day 7 (*p* = 0.090) and day 20 (*p* = 0.081). Furthermore, the lactose percentage of milk on day 7 (*p* = 0.056) tended to increase by sow’s HXT supplementation. Moreover, the sows that received VE at 100 mg/kg during late gestation and lactation increased α-tocopherol concentrations mainly in colostrum (*p* = 0.006), whereas accumulation of this compound was not as marked in day 7 milk (*p* = 0.062) and no changes were found on day 20. Dietary supplementation of HXT to sows from day 85 of pregnancy to day 28 of lactation also tended to increase (*p* = 0.093) the α-tocopherol concentration in colostrum and produced a marked increase in the day 7 milk retinol (vitamin A) concentration (*p* = 0.010).

No changes were observed by the combined supplementation of both antioxidants, except for the α-tocopherol concentration of colostrum that tended (*p* = 0.058) to be the greatest in the VE+HXT group than when VE was administered individually without HXT.

The estimated mean milk energy output (EMe) increased in the VE-supplemented groups (*p* = 0.0005), whereas HXT supplementation did not modify this parameter (Figure 1).

### 3.2. Fatty Acid Composition of Colostrum and Milk

The fatty acid composition of colostrum and milk on days 7 and 20 of lactation are presented in Table 2 and Table 3 and Figure 2.

The colostrum from sows supplemented with 100 mg/kg VE showed a greater proportion of C18:1 n−7 (*p* = 0.046) and tended to have a greater (*p* = 0.056) elongase C16 to C14 index than those groups that received 30 mg/kg VE (Table 2). The colostrum was also modified to a greater extent by the supplementation of 1.5 mg/kg HXT in the sow’s diets. Therefore, the HXT-supplemented groups had greater proportions of C18:2 n−6 (*p* = 0.002), C18:3 n−3 (*p* = 0.002), sum of total polyunsaturated fatty acids (∑PUFAs) (*p* = 0.002), ∑n−6 PUFAs (*p* = 0.002) and ∑n−3 PUFAs (*p* = 0.008) and lesser proportions of C18:1 n−9 (*p* = 0.027) and C20:1 n−9 (*p* = 0.044) than those groups without HXT supplementation.

EMe was also positively and linearly correlated with the ratio C18:1n−9/C18:0 of colostrum (r = 0.62, *p* = 0.0003), C16:1n−7/C16:0 (r = 0.71, *p* = 0.0001) (Figure 1), and in general with the Δ-9 and Δ-6 desaturase capacities (r = 0.52, *p* = 0.005 and r = 0.48, *p* = 0.012, respectively) of the sow.

Changes in the fatty acid proportion of milk on day 7 of lactation were also observed by dietary antioxidants supplementation, mainly in the VE-supplemented groups (Table 3). Hence, the administration of 100 mg/kg VE to sows from day 85 of pregnancy (VE effect) decreased C18:2 n−6 (*p* = 0.015), C18:3 n−3 (*p* = 0.011), ∑PUFAs (*p* = 0.028), ∑n−6 PUFAs (*p* = 0.027) and ∑n−3 PUFAs (*p* = 0.018) but increased C17:1 (*p* = 0.008), Δ-6-desaturase (*p* = 0.016) and Δ-5 + Δ-6-desaturase (*p* = 0.032) indices when compared with the groups that received 30 mg/kg VE. Moreover, C20:3 n−6 (*p* = 0.070), ∑PUFAs (*p* = 0.093) and ∑n−3 PUFAs (*p* = 0.062) tended to be lower in milk on day 7 when the sows were given the HXT supplementation. The combined administration of both antioxidants also modified the proportion of fatty acids in milk on day 7 of lactation in a different way (interaction effect). Thus, when VE was supplemented with HXT in the sow’s diets, this produced a significant decrease in C18:2 n−6 (*p* = 0.034), C18:3 n−6 (*p* = 0.018), C20:3 n−6 (*p* = 0.011), C20:4 n−6 (*p* = 0.030), ∑PUFAs (*p* = 0.013), ∑n−6 (*p* = 0.018), ∑n−3 (*p* = 0.014) and Δ-6 desaturase (*p* = 0.023) of milk than when VE was administered individually. The desaturase indices were not statistically affected on day 7 (*p* > 0.05) when both antioxidants were combined at high doses in the diets.

On day 20 of lactation, changes in the fatty acid profile of the milk were not as marked as in the milk on day 7. The milk from sows supplemented with HXT tended (*p* = 0.096) to have a greater C18:2 n−6 proportion when compared with the milk from non-HXT-supplemented sows but no significant changes were observed by the individual administration of VE (*p* > 0.05).

The combined administration of both antioxidants modified the composition of day 20 milk in a different way than that of day 7 milk. Hence, sows given VE+HXT produced milk with a greater proportion of C18:2 n−6 (*p* = 0.024), C18:3 n−3 (*p* = 0.047), ∑PUFAs (*p* = 0.029) and ∑n−6 PUFAs (*p* = 0.015) than when VE was supplemented individually in the diet (Figure 2). Moreover, the Δ-5 and Δ-6 desaturase indices were lower in the VE+HXT group than when VE was supplemented without HXT (*p* < 0.05).

### 3.3. Oxidative Stability of Milk and Oxidative Status of Sows and Piglets

The iron-induced oxidation of milk on days 7 and 20 of lactation is presented in Table 4. Sows given a VE-supplemented diet at 100 mg/kg produced milk that was less susceptible to oxidation on days 7 and 20 of lactation. Hence, the milk collected on day 7 of lactation from the VE-supplemented groups had lower malondialdehyde (MDA) concentrations after 30 min of incubation in the presence of ferrous sulfate and total MDA (*p* = 0.001) than those groups that received 30 mg/kg of VE. However, HXT supplementation did not statistically modify the lipid stability of milk on day 7 of lactation (*p* > 0.05) but decreased the stability of milk collected on day 20 (*p* = 0.028). Thus, milk from these HXT-supplemented groups had greater MDA concentrations than groups without HXT supplementation.

Moderate linear and positive correlations (Table 5) were found between the total MDA concentration of milk on day 7 and their composition in unsaturated fatty acids, mainly C18:2 n−6 (r = 0.39, *p* = 0.034), C18:3 n−3 (r = 0.42, *p* = 0.021) and ∑PUFAs (r = 0.36, *p* = 0.050). In addition, high linear and positive correlations were also observed between the MDA of milk on day 20 of lactation and the C18:2 n−6 proportion (r = 0.65, *p* = 0.001) and ∑PUFAs (r = 0.59, *p* = 0.002).

The plasma catalase activity of the sows was not statistically affected (*p* > 0.05), whereas the plasma α-tocopherol concentration was increased by the dietary VE supplementation (*p* = 0.0001). Moreover, lipid oxidation of the milk on day 20 was negatively correlated with the sow’s plasma catalase (r = −0.58, *p* = 0.017) and sow’s plasma α-tocopherol concentration on day 20 (r = −0.67, *p* = 0.006) (Table 5). Finally, the MDA concentration of milk on days 7 and 20 correlated to a great extent with the oxidative status of piglets (MDA) after weaning (r = 0.72, *p* = 0.003 and r = 0.73, *p* = 0.005, respectively) (Table 5).

## 4. Discussion

Milk is the main source of nutrients for a piglet before weaning and can affect its gut health and growth rate [4]. Therefore, in the present study, the use of antioxidants in sows’ diets (vitamin E and/or hydroxytyrosol) that could produce changes at the metabolic level [12,16] was evaluated in order to know whether their individual or combined dietary administration could modify the composition and lipid stability of the milk through lactation.

The composition of lactose, fat, and protein of colostrum and milk at different times of lactation were within the ranges described in the literature for Iberian sows [3] and pigs with improved genotypes [5,34]. As described by the latter authors, colostrum had a lower content of lactose and fat than milk, in agreement with the results of the present study. The evolution in protein composition could not be evaluated because the protein was not quantified in colostrum due to an insufficient sample. Dietary VE or HXT given to sows from day 85 of gestation to day 28 of lactation did not hardly modify the general composition of colostrum or milk on day 7 or 20. Rosales et al. [35] found that VE supplementation to ewes during late gestation and early lactation increased the lactose concentration of colostrum and fat concentration of milk, although the protein and lactose in milk did not differ between treatments. Moreover, Wang et al. [6] observed that milk from sows supplemented with 250 IU VE/kg feed during the last week of gestation and lactation had a greater content of fat. In contrast, the fat content of milk was not affected in the present study, which could have been due to the VE supplementation dose that was half of that used by Wang et al. [6], or in part to a smaller litter size in the case of the Iberian sow than in the white genotype used in Wang study (11 piglets born alive). Hence, after supplementing 90 IU/kg of VE to five-parity Landrace × Yorkshire sows, Chen et al. [36] did not find any effect on the 11-day milk composition. On the other hand, a concomitant fat increase was observed when milk protein decreased [34]; however, despite the decreasing trends in protein observed in the present study, no significant increases in fat content were observed.

Concerning the dietary HXT supplementation, either alone or in combination with VE, there is a lack of literature reporting the effects on general colostrum and milk composition. According to our findings, the general composition was not modified by the HXT in any case.

The concentrations of α-tocopherol and retinol in colostrum or milk were within the expected values according to dietary intervention [10,11,36,37,38] and decreased with lactation length, in agreement with previous research in the literature [11]. Dietary treatments modified the vitamin concentration of milk to different extents. Hence, dietary VE at 100 mg/kg feed increased the α-tocopherol in colostrum but did not modify the concentration in milk. In the present study, differences in supplementation dose (30 mg/kg vs. 100 mg/kg) and consequently VE accumulation were not as marked as in other investigations in the literature [39], in which the VE content was 2–3-fold above in milk samples from supplemented sows (200 mg/kg and 400 mg/kg VE) when compared with those that were non-supplemented (36 mg/kg VE). According to Mahan et al. [40] and Pinelly-Saavedra et al. [39], colostrum and milk α-tocopherol increased as dietary VE increased. In contrast, dietary HXT supplementation to sows did not modify the concentration of α-tocopherol of the colostrum or milk but increased retinol concentration on day 7 of lactation. To our knowledge, there were no previous studies on the possible effects of HXT on the vitamin composition of sow’s milk. In lactating ruminants, Terramoccia et al. [15] reported increased tocopherols and retinol concentrations in milk from buffaloes supplemented with dried stoned olive pomace. In addition, it was reported in rats that olive polyphenols may modify the metabolism of retinol and other lipid components and consequently have positive effects on some diseases [41].

The supplementation of both antioxidants did not modify the composition of the colostrum and milk. Bars-Cortina et al. [42] reported that a diet supplemented with some phenolic compounds of olive and thyme increased α-tocopherol in the tissues of rats. The combined administration of both antioxidants could produce a protective effect from their use or provide other antioxidant compounds [42]. However, other studies on pigs did not find any effect of the combination of VE and an olive-derived extract [16]. Discrepancies found in the literature can be explained by differences in the antioxidant source and/or supplementation dose.

The fatty acid proportions of colostrum and milk were within the values reported by other authors in sows [12,38,43]. Supplementation with VE in a sow’s diet only produced limited effects on the fatty acid composition of colostrum, as was observed by other authors in the literature [12]. Moreover, it should be noted that sows supplemented with VE during pregnancy and lactation allocate high proportions of C18:1n−7 to colostrum. This fatty acid and those of its n−7 group, such as C16:1 n−7, are easily β-oxidized to obtain energy [14], which may be an especially interesting aspect during the first hours of the piglet’s life. It is important to highlight that VE supplementation to sow’s diets produced more marked effects on milk at day 7 of lactation with a decrease in the C18-polyunsaturated fatty acids and ∑PUFAs, as well as a marked increase in desaturase indices of the n−6 and n−3 series that could, in part, explain the observed decrease in polyunsaturated C18 proportions. Then, when VE was given to sows independently, milk had the highest proportion of long PUFAs (C20:3 n−6 and C20:4 n−6) on day 7. A direct relationship between VE levels and the activity of the desaturase and elongase enzymes was described in the literature [44,45]. Thus, the lower the VE levels, the lower the activity of these enzymes. This fact was explained by the possible antioxidant effect of VE on desaturase and elongase enzymes [12,44,46], although a possible protective antioxidant effect on long-chain fatty acids was also postulated [46]. The changes in the desaturation index were maintained throughout lactation and were the highest when VE was given without HXT, although long PUFAs were not statistically modified at day 20 of lactation. These changes in the desaturation and, consequently, the different fatty acid classes imply that a piglet can better use them for metabolic purposes or other functions. Therefore, it was described that fatty acid is better metabolized when it has a higher number of unsaturations [47]. This is a relevant aspect in the particular case of the Iberian pig in which a reduced energy efficiency for growth and higher energy cost of body fat deposition during suckling was found [3,43]. In addition, a different transfer of long-chain fatty acids through milk could affect the composition of the tissue membrane or intestinal epithelium of the piglet [8] and affect the membrane fluidity or cell signaling [48] and, consequently, the animal’s health.

HXT supplementation of the sows’ diets also produced interesting changes in the fatty acid profile of colostrum and, to a lesser extent, in milk. The colostrum from HXT-supplemented groups had the highest content of ∑n−6 and ∑n−3 PUFAs that resulted in lower proportions of ∑MUFAs, mainly C18:1 n−9 and C20:1 n−9, and lower desaturase indices of the n−6 series. According to the results of the present study, the sows supplemented with HXT allocated a higher proportion of unsaturated fatty acids on the day of farrowing than the rest of the groups; however, after farrowing this potent ability to derive PUFAs to milk would be reduced. It was reported that olive-derived extracts increase glucose absorption [17] and help to produce faster fatty acid mobilization for different purposes [16]; in the specific case of a lactating sow, this could be used to address the energy needs for lactation and colostrum formation. However, it is of interest to observe the fact that the sows supplemented with HXT allocated many PUFAs to the colostrum, which may have resulted in a decrease in the desaturation indices and possibly a lower capacity of the sow to increase long PUFAs. Furthermore, a high proportion of C18-PUFAs in colostrum might result in a reduced capacity of the piglet to desaturate their long-chain derivatives, which, as reported before, are better utilized for energy purposes [43,47]; however, more research is needed. The fact of having less Δ-6 and Δ-5 desaturase activity should be taken into consideration since alterations of the Δ-5/Δ-6 activity have been associated with several diseases, from inflammation to tumorigenesis [49], and the modulation of Δ-5/Δ-6 activity could be considered as a possible therapeutic application [49].

In addition, although no significant change in the general milk composition was observed, it is interesting to point out that the transfer of nutrients to milk estimated as EMe output, that is, as a function of the litter size and litter weight, was not affected in the HXT groups but increased with VE supplementation. The higher desaturase capacity found in sows supplemented with VE could have preserved the utilization of PUFAs by the mother. Hence, Lauridsen and Danielsen [38] reported that a higher metabolic use and mobilization of C18:2 during lactation was related to lower fat and energy in milk, whereas milk from sows receiving MUFAs or SAT-enriched diets and lower PUFAs contained higher fat and energy. This fact was also observed in the present study in which analysis of data by multiple regression procedures indicated that 28% of the variation in EMe was explained by the proportion of C18:0 in colostrum and, consequently, with the desaturase capacity of the sow at the initial stage of lactation. Similarly, this would explain the positive correlation found between the desaturase capacity (C18:1/C18:0 and C16:1/C16:0) and the EMe output observed in the present research.

The supplementation of these different antioxidants in the diets of sows causes the mother to use different strategies to be able to promote the development of the piglet by diverting different fatty acids to the lactating gland. This could imply that the use of one or the other may affect gut health or favor the growth of the piglet at different times during lactation, depending on the specific fatty acid profile or milk composition. However, the use of the combination of both antioxidants during the first week of lactation did not seem to have beneficial effects on the fatty acid profile of the colostrum or milk compared with the independent use of each of them. In particular, the results of the supplementation with VE+HXT antioxidants implied a decrease in the derivation of unsaturated fatty acids from the n−6, n−3 or n−9 series toward milk on day 7, as well as a lower activity on the desaturation or elongation capacity of the sow. However, as lactation progressed and close to weaning, the combination of both antioxidants in the sows’ diet increased the milk PUFA content, especially in the C18 n−6 series. It would be expected that PUFA transfer decreased with the advancing of lactation [50] since C18:2 n−6 followed by the n−7 fatty acid series could be one of the most easily used for energy utilization by the mother [14]. This increase in PUFA enrichment in milk with the use of both antioxidants in the sow’s diet could, in part, be related to the decrease in the ability of the sow to desaturate fatty acids and obtain their long-chain derivatives close to the weaning time, which could also result in lower efficiency of milk energy for piglet growth [43]. However, more research is needed to clarify the possible effects on piglet growth or gut health.

The lipid stability of milk was measured in order to evaluate the oxidative capacity of the tested antioxidants and, consequently, the possible transfer to piglets of derivatives of the oxidation or other substances that may contribute to the control of oxidative stress. The groups supplemented with vitamin E had the lowest production of MDA in milk when compared with the other groups, whereas HXT supplementation resulted in an increase in oxidative-derived products. The antioxidant effects of vitamin E as a radical scavenger [7] and its capacity to protect against lipid oxidation were widely documented [7,10,11,32]. However, the higher range of oxidation values in milk from HXT groups could have been due to the high proportion of PUFAs, mainly on day 20 of lactation. Hence, in the present study, the total MDA production of milk was mainly explained by their PUFA content, as confirmed by the significant linear and positive correlation found between these components. It was reported that the more unsaturated the fatty acid chain, the higher the degree of oxidation [32]. In addition, it is of interest to point out that milk stability was also correlated with the oxidative status of the sow and their plasma tocopherol concentration. Data are only presented for day 20 because the plasma oxidative status of sows was not measured on day 7 of lactation, but similar results would be expected for the whole period. This is because, in the present research, a high and positive correlation between the milk MDA concentration and the piglet’s oxidative status was also detected at different lactation stages. The fact that the piglet’s plasma MDA concentration (as a derivative of oxidation and measurement of oxidative stress) was positive and highly correlated with the total MDA concentration of milk could indicate the high importance of milk as a main vehicle for different components, not only the antioxidant substances from the mother but also those resulting from the different metabolic state of the sow that determines a different proportion of fatty acids during the process of milk synthesis in the sow’s mammary gland. This is of relevance for those nutrients with limited placental transfer, such as vitamin E [9], which are very important for not only maintaining the oxidative status but also those that provide energy during the first weeks of life.

## 5. Conclusions

In conclusion, the supplementation of sows with VE or HXT slightly modified the general composition of milk in different ways. The changes were more marked in the fatty acid profile. VE produced an increase in the n−7 of the colostrum and long PUFAs and Δ-5 and Δ-6 indices of the milk during lactation, as well as the lower production of TBARs products. Nevertheless, HXT supplementation markedly increased the C18-PUFAs of the colostrum and moderately so as lactation progressed with the decrease in the sow’s ability to desaturate. Maternal VE supplementation produced a more beneficial milk composition to improve the oxidative status of piglets, which could promote gut health and the piglet’s growth during the first weeks of life; however, further studies are needed to clarify this.

## Figures and Tables

**Figure 1 antioxidants-12-01039-f001:**
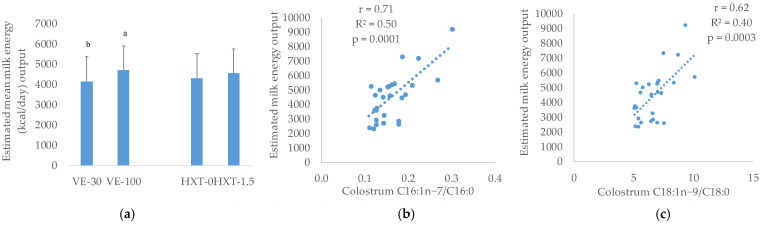
Estimated mean milk energy output (Kcal/day) (EMe) during lactation from sows given α-tocopherol (VE: 30 vs. 100 mg/kg) or hydroxytyroxol (HXT: 0 vs. 1.5 mg/kg) from day 85 of gestation (**a**); correlations between EMe and desaturase capacity of sows measured as C16:1n−7/C16:0 (**b**) and C18:1n−9/C18:0 (**c**) in colostrum. ^a,b^ Different superscript letters signify a statistically significant difference; *p*—differences were statistically different when *p* < 0.05.

**Figure 2 antioxidants-12-01039-f002:**
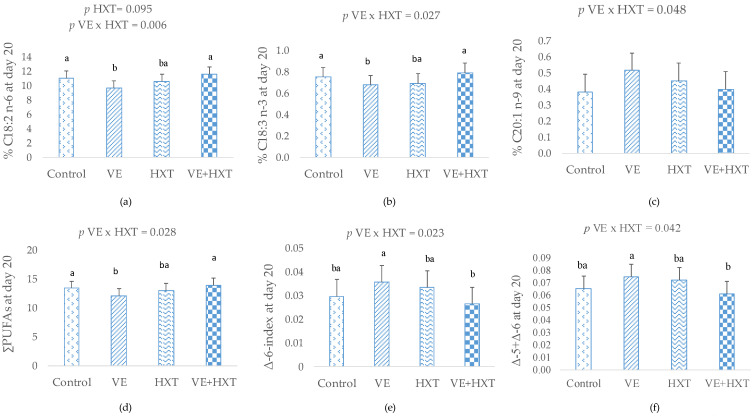
Interaction effect of dietary antioxidant α-tocopherol (VE: 30 vs. 100 mg/kg), hydroxytyroxol (HXT: 0 vs. 1.5 mg/kg) or their combination provided to sows from day 85 of gestation to weaning on the fatty acid composition (%) of milk on day 20 of lactation: (**a**) % C18:2 n−6; (**b**) % C18:3 n−3; (**c**) % C20:1 n−9; (**d**) ∑PUFAs—sum of total polyunsaturated fatty acids; (**e**,**f**) Δ-5 = (C20:4n−6)/(C20:4n−6 + C20:3n−6); Δ-6 = (C18:3n−6 + C18:4n−3)/(C18:2n−6 + C18:3n−3 + C18:3n−6 + C18:4n−3); ^a,b^ Different superscript letters signify a statistically significant difference; *p*—differences were statistically different when *p* < 0.05.

**Table 1 antioxidants-12-01039-t001:** General composition of colostrum and milk from sows given different levels of α-tocopherol (VE) or hydroxytyroxol (HXT) from day 85 of gestation.

	Control	VE	HXT	VE+HXT	RMSE ^1^	*p* VE ^2^	*p* HXT	*p* VE × HXT
*Colostrum*								
Dry matter, %	21.60	20.64	22.40	22.39	3.981	0.760	0.466	0.793
Lactose, %	3.64	5.04	3.91	4.32	1.437	0.090	0.728	0.399
Fat, %	5.55	5.39	5.98	6.97	2.504	0.938	0.498	0.810
Tocopherol, µg/mL	13.44 ^b^	15.65 ^ab^	12.90 ^b^	25.62 ^a^	5.972	0.006	0.093	0.058
Retinol, µg/mL	0.83	0.80	0.68	1.46	0.589	0.200	0.439	0.169
*Milk day 7*								
Dry matter, %	18.06	16.05	17.33	17.16	3.957	0.445	0.893	0.518
Lactose, %	5.36 ^ab^	5.23 ^b^	6.51 ^ab^	6.48 ^a^	1.696	0.896	0.056	0.942
Fat, %	7.37	6.40	7.49	9.08	2.359	0.721	0.113	0.144
Protein, %	5.59 ^a^	4.89 ^b^	5.00 ^ab^	4.91 ^b^	0.632	0.091	0.226	0.196
Tocopherol, µg/mL	3.49 ^b^	3.95 ^ab^	3.50^b^	4.25 ^a^	0.888	0.062	0.641	0.641
Retinol, µg/mL	0.25 ^b^	0.33 ^ab^	0.46 ^ab^	0.52 ^a^	0.202	0.346	0.010	0.911
*Milk day 20*								
Dry matter, %	17.37	16.91	17.94	17.10	1.879	0.418	0.637	0.811
Lactose, %	6.05	5.21	7.99	6.06	2.106	0.105	0.101	0.515
Fat, %	7.47	7.57	7.97	7.55	2.257	0.859	0.784	0.773
Protein, mg/mL	5.35 ^ab^	5.11 ^b^	5.87 ^a^	5.30 ^ab^	0.563	0.081	0.125	0.449
Tocopherol, µg/mL	2.90	3.32	2.44	2.84	0.737	0.167	0.117	0.966
Retinol, µg/mL	0.39	0.48	0.48	0.55	0.122	0.415	0.421	0.891

^1^ RMSE—root-mean-square error (pooled SD) (n = 14 replicates for each main effect, n = 7 replicates for the interaction); ^2^ *p*—differences were statistically different when *p* < 0.05. ^a,b^ Different superscript letters signify a statistically significant difference.

**Table 2 antioxidants-12-01039-t002:** Fatty acid composition (%) of colostrum from sows given different levels of α-tocopherol (VE) or hydroxytyroxol (HXT) from day 85 of gestation.

% Fatty Acids	Control	VE	HXT	VE+HXT	RMSE ^12^	*p* VE ^13^	*p* HXT	*p* VE × HXT
C14:0	1.25	1.15	1.27	1.22	0.139	0.180	0.418	0.678
C14:1	0.04	0.03	0.04	0.02	0.021	0.146	0.240	0.796
C16:0	22.01	22.27	22.12	23.37	1.449	0.328	0.470	0.601
C16:1n−9	1.26	1.20	1.30	1.21	0.169	0.446	0.395	0.847
C16:1n−7	3.70	3.89	3.48	3.11	0.873	0.916	0.275	0.636
C17:0	1.10	1.07	1.04	0.96	0.306	0.652	0.495	0.851
C17:1	0.46	0.47	0.49	0.50	0.068	0.982	0.433	0.807
C18:0	6.27	6.43	6.06	6.75	0.972	0.580	0.676	0.889
C18:1n−9	41.44	41.84	39.76	38.70	2.521	0.836	0.027	0.542
C18:1n−7	3.05	3.58	3.14	3.15	0.608	0.046	0.829	0.848
C18:2n−6	15.76 ^ab^	14.58 ^b^	17.52 ^a^	17.53 ^a^	1.643	0.225	0.002	0.529
C18:3n−6	0.23	0.22	0.21	0.17	0.059	0.659	0.481	0.995
C18:3n−3	0.94 ^ab^	0.82 ^b^	1.10 ^a^	1.09 ^a^	0.142	0.100	0.002	0.583
C18:4n−3	0.20	0.18	0.18	0.17	0.063	0.671	0.830	0.732
C20:0	0.09	0.08	0.07	0.05	0.041	0.540	0.494	0.790
C20:1n−9	0.34	0.39	0.29	0.25	0.092	0.466	0.044	0.511
C20:2	0.48	0.50	0.46	0.45	0.079	0.388	0.558	0.938
C20:3n−6	0.24	0.21	0.23	0.23	0.044	0.773	0.490	0.126
C20:4n−6	1.13	1.09	1.22	1.06	0.198	0.611	0.239	0.979
∑SAT ^1^	30.72	30.99	30.57	32.36	2.276	0.479	0.819	0.691
∑MUFAs ^2^	50.30	51.41	48.51	46.94	3.410	0.828	0.057	0.535
∑PUFAs ^3^	18.99 ^ab^	17.60 ^b^	20.92 ^a^	20.70 ^a^	1.783	0.193	0.002	0.496
∑n−6 ^4^	17.37 ^ab^	16.10 ^b^	19.18 ^a^	18.99 ^a^	1.649	0.198	0.002	0.502
∑n−3 ^5^	1.14 ^ab^	1.00 ^b^	1.28 ^a^	1.25 ^a^	0.162	0.109	0.008	0.540
Δ-9-desaturase ^6^	0.61	0.61	0.60	0.57	0.033	0.621	0.199	0.571
Δ-5-desaturase ^7^	0.83	0.84	0.84	0.82	0.019	0.828	0.819	0.056
Δ-6-desaturase ^8^	0.03	0.03	0.02	0.02	0.006	0.923	0.065	0.830
Thioesterase (16-14) ^9^	0.95	0.95	0.95	0.95	0.006	0.067	0.759	0.943
Elongase (18-16) ^10^	0.22	0.22	0.21	0.22	0.021	0.816	0.385	0.922
Elongase (20-18) ^11^	0.01	0.01	0.01	0.01	0.007	0.529	0.757	0.765

^1^ ∑SAT—sum of total saturated fatty acids; ^2^ ∑MUFAs—sum of total monounsaturated fatty acids; ^3^ ∑PUFAs—sum of total polyunsaturated fatty acids; ^4^ ∑n−6—sum of total n−6 fatty acids; ^5^ ∑n−3—sum of total n−3 fatty acids; ^6^ Δ-9−desaturase index = (C14:1 + C16:1 + C18:1)/C14:0 + C14:1 + C16:0 + C16:1 + C18:0 + C18:1); ^7^ Δ-5-desaturase = (C20:4n−6)/(C20:4n−6 + C20:3n−6); ^8^ Δ-6-desaturase = (C18:3n−6 + C18:4n−3)/(C18:2n−6 + C18:3n−3 + C18:3n−6 + C18:4n−3); ^9^ thioesterase index = C16:0/C14:0; ^10^ elongase (18/16) index = C18:0/C16:0; ^11^ elongase (20/18) index = C20:0/C18:0; ^12^ RMSE—root-mean-square error (pooled SD) (n = 14 replicates for each main effect, n = 7 replicates for the interaction); ^13^
*p*—differences were statistically different when *p* < 0.05; ^a,b^ Different superscript letters signify a statistically significant difference.

**Table 3 antioxidants-12-01039-t003:** Fatty acid composition (%) of milk on day 7 of lactation from sows given different levels of α-tocopherol or hydroxytyroxol from day 85 of gestation.

% Fatty Acids	Control	VE	HXT	VE+HXT	RMSE ^12^	*p* VE ^13^	*p* HXT	*p* VE × HXT
C14:0	3.08	2.77	2.96	2.90	0.544	0.370	0.981	0.534
C14:1	0.19	0.15	0.19	0.17	0.058	0.168	0.777	0.763
C16:0	29.81	28.29	28.07	28.55	3.049	0.639	0.507	0.372
C16:1n−9	0.37	0.54	0.42	0.40	0.189	0.294	0.530	0.177
C16:1n−7	7.82	7.01	7.57	7.13	1.653	0.305	0.916	0.756
C17:0	0.71	0.90	0.78	0.72	0.195	0.358	0.417	0.076
C17:1	0.41 ^b^	0.52 ^ab^	0.46 ^ab^	0.56 ^a^	0.096	0.008	0.205	0.896
C18:0	4.62	4.92	4.45	5.35	0.877	0.071	0.694	0.351
C18:1n−9	34.83	35.86	35.78	38.02	3.545	0.212	0.235	0.636
C18:1n−7	1.85	2.44	2.42	2.39	0.515	0.149	0.170	0.111
C18:2n−6	13.55 ^a^	13.35 ^a^	13.97 ^a^	11.19 ^b^	1.594	0.015	0.143	0.034
C18:3n−6	0.10 ^b^	0.15 ^a^	0.13 ^b^	0.12 ^b^	0.028	0.094	0.700	0.018
C18:3n−3	0.97 ^a^	0.94 ^a^	0.99 ^a^	0.78 ^b^	0.120	0.011	0.135	0.054
C18:4n−3	0.18	0.22	0.19	0.18	0.042	0.390	0.343	0.149
C20:0	0.10	0.12	0.10	0.09	0.044	0.573	0.285	0.302
C20:1n−9	0.30	0.38	0.34	0.37	0.108	0.160	0.762	0.576
C20:2	0.35	0.44	0.40	0.40	0.091	0.239	0.916	0.192
C20:3n−6	0.14 ^b^	0.22 ^a^	0.16 ^b^	0.13 ^b^	0.052	0.285	0.071	0.012
C20:4n−6	0.60 ^b^	0.80 ^a^	0.64 ^ab^	0.56 ^b^	0.166	0.288	0.101	0.029
∑SAT ^1^	38.31	37.01	36.35	37.60	3.300	0.980	0.570	0.293
∑MUFAs ^2^	45.78	46.89	47.17	49.04	3.128	0.199	0.128	0.740
∑PUFAs ^3^	15.91 ^a^	16.11 ^a^	16.48 ^a^	13.36 ^b^	1.731	0.028	0.093	0.013
∑n−6 ^4^	14.40 ^a^	14.52 ^a^	14.90 ^a^	12.00 ^b^	1.650	0.027	0.102	0.018
∑n−3 ^5^	1.15 ^a^	1.16 ^a^	1.18 ^a^	0.96 ^b^	0.116	0.018	0.062	0.014
Δ-9-desaturase ^6^	0.53	0.55	0.55	0.55	0.035	0.625	0.326	0.683
Δ-5-desaturase ^7^	0.80	0.79	0.80	0.81	0.048	0.945	0.675	0.498
Δ-6-desaturase ^8^	0.02 ^b^	0.03 ^a^	0.02 ^ab^	0.02 ^ab^	0.005	0.016	0.883	0.427
Δ-5+Δ-6-desaturase	0.05 ^b^	0.07 ^a^	0.05 ^b^	0.05 ^ab^	0.013	0.032	0.279	0.135
Thioesterase (16-14) ^9^	0.91	0.91	0.90	0.91	0.011	0.255	0.597	0.959
Elongase (18-16) ^10^	0.14	0.15	0.14	0.16	0.031	0.096	0.579	0.662
Elongase (20-18) ^11^	0.02	0.02	0.02	0.02	0.008	0.917	0.173	0.157

^1^ ∑SAT—sum of total saturated fatty acids; ^2^ ∑MUFAs—sum of total monounsaturated fatty acids; ^3^ ∑PUFAs—sum of total polyunsaturated fatty acids; ^4^ ∑n−6—sum of total n−6 fatty acids; ^5^ ∑n−3—sum of total n−3 fatty acids; ^6^ Δ-9−desaturase index = (C14:1 + C16:1 + C18:1)/C14:0 + C14:1 + C16:0 + C16:1 + C18:0 + C18:1); ^7^ Δ-5-desaturase = (C20:4n−6)/(C20:4n−6 + C20:3n−6); ^8^ Δ-6-desaturase = (C18:3n−6 + C18:4n−3)/(C18:2n−6 + C18:3n−3 + C18:3n−6 + C18:4n−3); ^9^ thioesterase index = C16:0/C14:0; ^10^ elongase (18/16) index = C18:0/C16:0; ^11^ elongase (20/18) index = C20:0/C18:0; ^12^ RMSE—root-mean-square error (pooled SD) (n = 14 replicates for each main effect, n = 7 replicates for the interaction); ^13^
*p*—differences were statistically different when *p* < 0.05; ^a,b^ Different superscript letters signify a statistically significant difference.

**Table 4 antioxidants-12-01039-t004:** Iron-induced lipid oxidation (nmols MDA/mg protein) of milk on days 7 and 20 of lactation from sows given α-tocopherol (VE: 30 vs. 100 mg/kg) or hydroxytyroxol (HXT: 0 vs. 1.5 mg/kg) from day 85 or gestation.

	Control	VE	HXT	VE+HXT	RMSE ^1^	*p* VE ^2^	*p* HXT	*p* VE × HXT
*Milk day 7, nmols MDA/mg protein*								
MDA 0 min	0.15 ^a,b^	0.14 ^b^	0.17 ^a^	0.15 ^ab^	0.024	0.134	0.075	0.263
MDA 30 min	0.14	0.10	0.14	0.12	0.063	0.183	0.713	0.664
MDA 90 min	0.61 ^a^	0.28 ^b^	0.42 ^ab^	0.25 ^b^	0.188	0.001	0.126	0.240
MDA 120 min	0.30 ^a^	0.11 ^b^	0.31	0.16 ^ab^	0.164	0.010	0.592	0.770
Total ∑MDA ^3^	1.20 ^a^	0.56 ^c^	1.05 ^ab^	0.69 ^bc^	0.368	0.001	0.932	0.298
*Milk day 20, nmols MDA/mg protein*								
MDA 0 min	0.15	0.12	0.14	0.14	0.037	0.317	0.760	0.253
MDA 30 min	0.11 ^b^	0.10 ^b^	0.24 ^a^	0.13 ^b^	0.091	0.097	0.039	0.182
MDA 90 min	0.22 ^b^	0.15 ^b^	0.48 ^a^	0.23 ^b^	0.128	0.005	0.003	0.079
MDA 120 min	0.24	0.10	0.23	0.21	0.167	0.259	0.456	0.394
Total ∑MDA	0.72 ^ab^	0.48 ^b^	1.09 ^a^	0.72 ^ab^	0.328	0.026	0.028	0.604

^1^ RMSE—root-mean-square error (pooled SD) (n = 14 replicates for each main effect, n = 7 replicates for the interaction); ^2^ *p*—differences were statistically different when *p* < 0.05. ^3^ MDA—malondialdehyde concentration; ^a,b,c^ Different superscript letters signify a statistically significant difference.

**Table 5 antioxidants-12-01039-t005:** Linear equations between the total MDA concentration (nmols/mg protein) of milk at 7 or 20 days of lactation and milk fatty acid proportion, sow’s plasma oxidative status (catalase and α-tocopherol concentrations) or piglet’s oxidative status (MDA).

	Intercept	±s.d. ^1^	Slope	±s.d.	Variable x	r	R^2^	*p* (Linear) ^2^
*Milk on day 7*								
Total MDA^3^	−0.424 ± 0.62		0.107 ± 0.05		% C18:2 n−6 in milk	0.39	0.16	0.034
Total MDA	−0.486 ± 0.59		1.593 ± 0.65		% C18:3 n−3 in milk	0.42	0.18	0.021
Total MDA	−0.391 ± 0.68		0.088 ± 0.04		∑PUFAs in milk ^4^	0.36	0.13	0.050
Total MDA	0.525 ± 0.13		9.133 ± 2.44		MDA in piglets after weaning	0.72	0.52	0.003
*Milk on day 20*								
Total MDA	−1.298 ± 0.51		0.189 ± 0.05		% C18:2 n−6 in milk	0.65	0.43	0.001
Total MDA	−1.510 ± 0.66		0.173 ± 0.05		∑PUFAs in milk	0.59	0.35	0.002
Total MDA	2.683 ± 0.73		−0.097 ±0.04		Sow’s plasma catalase on day 20	–0.58	0.34	0.017
Total MDA	1.063 ± 0.13		−0.183 ± 0.06		Sow’s plasma α-tocopherol (µg/mL) on day 20	–0.67	0.45	0.006
Total MDA	0.344 ± 0.12		7.238 ± 2.06		MDA in piglets after weaning	0.73	0.53	0.005

^1^ s.d.—standard deviation of mean; ^2^ *p*—differences were statistically different when *p* < 0.05; ^3^ MDA—malondialdehyde concentration; ^4^ ∑PUFAs—sum of total polyunsaturated fatty acids.

## Data Availability

The data are contained within this article.

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
