# Peer review of "Dietary Vitamin E and/or Hydroxytyrosol Supplementation to Sows during Late Pregnancy and Lactation Modifies the Lipid Composition of Colostrum and Milk"

_antioxidants, 2023, doi:10.3390/antiox12051039_

Round 1

Reviewer 1 Report

Dear authors,

Let me start by congratulating you on the article. I found the study to be complex and contain new valuable and interesting information’s.

The Abstract and the Introduction are appropriate and the aim of the work clearly established.

Both the experimental design and the presentation are logical.

Regarding the methodology used, I appreciated first of all the very clear structure and the multitude of tests used.

The results and discussions are presented in a proper manner, with reference to recent literature data.

Author Response

Dear reviewer, many thanks for your positive comments regarding the paper

Reviewer 2 Report

The manuscript by Laviano H. D. and colleagues, entitled “Dietary vitamin E and/or hydroxytyrosol supplementation to sows during late pregnancy and lactation modifies composition of colostrum and milk”, reference number Antioxidants-2023-2319609, aims to alter the lipid composition of sow’s milk through supplementation of vitamin E and/or hydroxytyrosol as a successful strategy to improve the intestinal health and growth of the piglets during the first weeks of life. Although the subject is updated and the manuscript is reasonable written and sound, I must add that the supplementation with vitamin E is no longer new. On the contrary, the supplementation with hydroxytyrosol (HXT) is. Moreover, there are some relevant data missing. Apart from these general comments, along I was reading the manuscript, some important considerations and problems came up. And here they are, point by point:

 Major and minor comments:

 1.                  In the title of the manuscript, please add lipid composition (page 1, line 3).

 2.                  Please, replace antioxidant micronutrients by antioxidant micronutrients supplementation (page 2, line 64).

 3.                  I am confused… What is the relation between 100 mg/kg of vitamin E and 250 IU/kg above cited (page 2, line 48)?

 4.                  I think the topic of post-weaned piglets should be more developed in the Introduction.

 5.                  Why choosing day 7 and 20 days post-partum for collecting blood samples from sows (page 3, line 119)? The authors need to be more specific. Add some bibliographic references to support your choice, if necessary.

6.                  Only one male piglet was selected from each litter at weaning? I do not think it is enough for statistical power reasons (page 3, line 120). Would you like to comment?

 7.                  The determination of lactose was performed by a commercial kit (page 3, line 132)? Add some bibliographic references regarding the laboratory protocol.

 8.                  Please, replace milk was put by milk was placed (page 3, line 138).

 9.                  In page 3, line 121, blood was collected into sterile EDTA vacuum tubes and plasma samples were kept at -80ºC until analysis. Please add plasma samples.

10.              The D9 - desaturase index needs to be supported by bibliographic references (page 4 , line 172).

 11.              Please add the protocol for the determination of enzymatic activity of catalase in the Laboratory analysis description (page 4, lines 194-196) making use of some bibliographic references. It is missing.

 12.              Please explain in full detail the meaning of RMSE the first time it appears in the test (page 5, line 203).

 13.              The authors have chosen GLM from SAS to perform the statistical analysis. I wonder if Proc Mix (a mixed model) would not be better (page 5, line 200).

 14.              I do not believe in statistical trends (for example: page 5, line 214). “Whether it rains, or it does not rain”. I think the authors should remove all the statistical trends along the entire manuscript.

 15.              Why did the authors choose 100 mg/kg of vitamin E (page 5, line 217)? Have you based this value in the literature?

 16.              The authors should use significant numbers in the Tables. Moreover, p value contains 3 decimal points, not four (see for example Table 1, Table 2, Table 3, Table 4, Table 5).

 17.              Whenever using correlations, please keep in mind that: strong correlation if r > 0.7;  weak correlation if r < 0.3; moderate correlation in between those values (Costa P. et al. 2012. Effect of low- and high-forage diets on meat quality and fatty acid composition of Alentejana and Barrosã beef breeds. Animal 6: 1187-1197).

 18.              Table 4 footnote should include p < 0.05 when describing statistical significant differences (page 10).

 19.              r values and p values should have three decimal points (see Figure 1, page 6).

 20.              Please, consider the following sentence: “The estimated mean milk energy output (EMe) increased in VE supplemented groups (p < 0.05), whereas HXT supplementation did not modify this parameter (Figure 1) (page 6, lines 233-235). Add the exact p value found.

 21.              Figure 1 caption should include p < 0.05 when describing statistical significant differences (page 6).

 22.              Please, consider the following sentence: “Desaturases indices were not statistically affected at day 7, when both antioxidants were combined as high doses in the diets (page 9, lines 303-305). Please add p > 0.05 at the end of the sentence. The same applies for lines 308-309, page 9.

 23.              In Figure 2, please add a, b, c, d, e and f to the individual panels in order to facilitate the understanding of the plate.

 24.              Detail the interest in determining D5 and D6 desaturases indices supported by adequate bibliographic references.

 25.              “… but decreased stability of milk collected at day 20.” (page 10, line 331) Please add the exact p value.

 26.              “Hence, milk collected at day 7 of lactation from VE-supplemented groups had lower malondialdehyde (MDA) concentration…” (page 10, lines 327-328). Please, add the exact p value. The authors should be more precise.

 27.              “However, HXT supplementation did not statistically modify the lipid stability of milk at day 7 of lactation…” (page 10, lines 330-331). Please, add p > 0.05.

 28.              Where are the results for the enzymatic activity of catalase in plasma? (page 10, line 347 and page 11, Table 5)

 29.              Where are the results for the determination of alpha-tocopherol in plasma? (page 10, line 347 and page 11, Table 5)

 30.              Which parameters did the authors include to assess plasma oxidative status (see Table 5, page 11)? Are you referring to the enzymatic activity of catalase or alpha-tocopherol measurements in plasma? This is confusing.

 31.              There is no mention to catalase enzyme in the Discussion section (pages 11-14).

 32.              I think the Discussion is too long (page 11-14).

 33.              Also in the Discussion section, remove the statistical trends found (for example, page 12, lines 414-417).

 34.              Please, consider the following sentence: “To the best of our knowledge, there is no information in the literature that reports any effect of HXT supplementation to sows on the fatty acid composition of colostrum and milk, so this is the first study that provides data” (page 13, lines 457-459). In my opinion, this sentence belongs to the Introduction. In fact, it should be placed at the End of Introduction.

 35.              Again the same for the sentence: “On the other hand, the effects of HXT on lipid stability of sow’s milk have not been studied previously” (page 14, lines 513-515).

 36.              In the Conclusions, the sentence: “Changes produced by VE supplementation could possibly be more beneficial to promote gut health and piglet’s growth during the first weeks of life…” (page 14, lines 542-543) is speculative. Please, rephrase this sentence.

 37.              The authors included 46 bibliographic references in the list of literature, some of which are recent and updated (2019, 2020 and 2021).

Reviewer 3 Report

The objective of the article antioxidants-2319609 (Dietary vitamin E and/or hydroxytyrosol supplementation to sows during late pregnancy and lactation modifies composition of colostrum and milk) was to investigate the effects of dietary supplementation of vitamin E, hydroxytyrosol or their combination to sows during gestation and lactation on colostrum and milk composition, as well as on piglet’s oxidative status.

This study deals with an interesting topic and fits with the journal's scope. The results are novel and provide new information for the growth and gut health of piglets.

I suggest the acceptance under a minor revision based on the following major and minor comments. 

Comments for the authors

Major comments

Introduction

-        Add a paragraph for the oxidative stress in newborn piglets and the consequences 

Materials and Methods

-        L95-110: provide details about the experimental animals (e.g., mean BW, age, parity distribution, vaccinations)

-        L115: hand-milking?? Did you collect 20 ml per sow with hand-milking using hygiene measures (sterile gloves, cleaning of the nipple and surrounding area of the sow with soap and sterile water, use of cotton soaked with 75% ethyl alcohol to minimize the contamination by skin bacteria)? 

Discussion:

-        L363-368: add appropriate references 

Minor comments

-        Title: Dietary vitamin E and/or hydroxytyrosol supplementation to sows during late pregnancy and lactation modifies the composition of colostrum and milk

-        L31: … the main effects..

-        L32:  between the estimated..

L34: … with the sow’s plasma oxidative status

-        L50: .. which has been shown....

-        L51-52: The health and productivity effect..

-        L72: … milk has not been investigated..

-        L155: A lyophilized sample

-        L386: .. the 11-day milk composition

-        L438: .. on day 7.

-        L450: … during suckling has been found

-        L461: .. fatty acids on the day of farrowing

-        L478: .. was related to lower fat

-        L483: .. capacity of the sow at the initial stage

-        L503: .. both antioxidants in the sow’s diet could be in part related to the decrease in the ability of the sow to desaturate fatty acids and obtain their long-chain derivatives close to the weaning time, which could also result in lower efficiency of milk energy for piglet growth

-        L515:.. on the lipid stability

Kind regards
